# Optimizing Materials with CliqueFlowmer

## Abstract

Recent advances in deep learning inspired computational approaches to enhance *material discovery* (MD). A plethora of problems in this field involve finding materials that optimize a target property. Nevertheless, the increasingly popular generative modeling-based methods are ineffective at boldly exploring attractive regions of the material space due to their maximal likelihood training. In this work, we offer an alternative MD technique based on offline *model-based optimization* (MBO) that fuses direct optimization of a target material property into generation. For that end, we introduce a domain-specific model, dubbed *CliqueFlowmer*, that incorporates recent advances of clique-based MBO into transformer and flow generation. We validate CliqueFlowmer's optimization abilities and show that materials it produces strongly outperform those provided by generative baselines.

## 1 Introduction

Large neural network models have recently enabled solving challenging artificial intelligence tasks such as language modeling, automated coding, and image generation (Achiam et al., 2023; Team et al., 2024; Ho et al., 2020; Esser et al., 2024). Meanwhile, scientific problems that deal with the world of atoms, rather than the world of bits, are yet to benefit from this revolution (Thiel, 2025). Indeed, exploration of new physical breakthroughs continues to take place in physical wet labs, and is driven by costly physical experimentation (Freese et al., 2024; Shahzad et al., 2024). One of such problems is *material discovery* (MD), wherein scientists strive to invent new chemical structures that display properties absent in known materials (Jain et al., 2013; Lin et al., 2025). A systematic, sample-efficient methodology for material discovery (MD) has the potential to deliver structures that serve as catalysts for key energy-conversion reactions and as structural and functional components of advanced biomaterials. Such advances would greatly accelerate the development of clean-energy technologies and medical therapies, effectively extending the recent AI-driven progress from the world of bits to the world of atoms (Gokcekuyu et al., 2024; Han & Su, 2025).

Having realized the importance of this problem, AI researchers have been increasingly turning their attention to AI-driven MD. Most commonly, the introduced methods have been utilizing the celebrated *diffusion* and *flow* models that had become very successful in the domain of image generation (Ho et al., 2020; Lipman et al., 2022; Miller et al., 2024; Inizan et al., 2025). While effective at harnessing the distribution of viable materials presented to them in the dataset, likelihood-based generative models do not actively explore the relation between the materials and their properties. This is a major impediment in settings where MD is expected to optimize materials with respect to a specified property (Yang et al., 2024; Havens et al., 2025). In the meantime, a new paradigm, known as *offline model-based optimization* (MBO) has made steps towards techniques that optimize *scientific designs* by bootstrapping models trained entirely on offline data (Kumar et al., 2021; Trabucco et al., 2022; Kuba et al., 2024). Nevertheless, MBO methods have been mainly deployed in standard benchmark tasks, free, for example, of the challenging intricacies of MD in which data is inherently of the hybrid discrete-continuous nature, irregular shape, and subject to physical constraints.

To address the need for AI-driven MD and the limitations of generative approaches, in this work, we introduce *CliqueFlowmer*—a model that renders material data tractable by MBO and enables optimizing material structures. The core of the model is an auto-encoder that converts the multi-modal, irregular material data into structured, finite-dimensional vectors that can be optimized with *clique-based* MBO (Kuba et al., 2024), and mapped back into the material form. These abilities are attained by a carefully engineered neural network architecture that utilizes transformers and combines them with generative learning techniques such as next-token prediction and flow matching,

and MBO techniques such as clique decomposition (Grudzien et al., 2024). Empirically, utilizing data from Materials Project (Jain et al., 2013) as well as M3GNet (Chen & Ong, 2022) and MEGNet (Chen & Ong, 2022) as oracles for the studied properties, we show that materials optimized by CliqueFlowmer vastly outperform those sampled by generative baselines. Additionally, we hope that this work will attract attention to the field of MBO and will open up possibilities for more effective MD methods.

## 2 BACKGROUND

This section provides the necessary background of the key notions discussed by our work. First, we lay down foundations of material modeling that enables us to use machine learning methods in MD. Then, we introduce the basic ideas of continuous normalizing flows (*i.e.,* flow matching). Lastly, we state the problem that we aim to solve (MD) and formulate it through the lens of offline MBO.

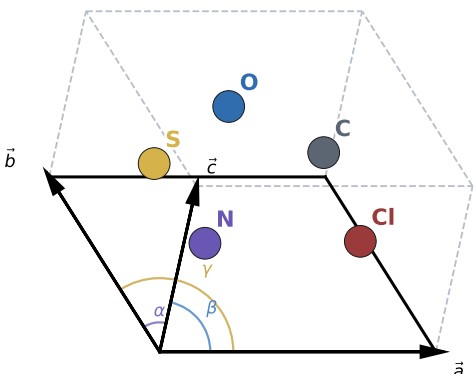

### 2.1 MODELING MATERIALS

We characterize a material M by its *unit cell—* the material's unit amount. The cell has a shape of a parallelopiped determined by the lengths of its axes, $(a, b, c) \in \mathbb{R}^3_+$, and angles between them, $(\alpha, \beta, \gamma) \in (0, \pi)^3$. The content of the cell is a set of $N_{\text{atom}}$ (which varies between materials) atoms that is represented by the sequence $\mathbf{a} \in \mathcal{A}^{N_{\text{atom}}}$ of their types, as well as their positions $\mathbf{X} \in [0, 1)^{N_{\text{atom}} \times 3}$ expressed in the basis induced by $(a, b, c)$ and $(\alpha, \beta, \gamma)$. Thus, the space of materials M can be embedded in the product $\mathcal{M} = \mathbb{R}^3_+ \times (0, \pi)^3 \times (\mathcal{A} \times [0, 1)^3)^*$, and a single material can be characterized as a tu-

Figure 1: The unit cell of a hypothetical material. The cell has a shape of a parallelopiped determined by three axes, $\vec{a}$, $\vec{b}$, and $\vec{c}$. The angles between the axes are $\text{ang}(\vec{b}, \vec{c}) = \alpha$, $\text{ang}(\vec{c}, \vec{a}) = \beta$, $\text{ang}(\vec{a}, \vec{b}) = \gamma$. In this cell, there are five atoms, whose type sequence is $\mathbf{a} = [\text{C, O, N, Cl, S}]$.

ple $[(a, b, c), (\alpha, \beta, \gamma), \mathbf{a}, \mathbf{X}]$. [1] In this paper, we denote the geometrical part of the material as $\mathbf{G} = [(a, b, c), (\alpha, \beta, \gamma), \mathbf{X}]$, and thus can write $[\mathbf{a}, \mathbf{G}]$ to represent material M.

### 2.2 FLOW MATCHING

The goal of a continuous normalizing flow (Lipman et al., 2022) is to learn sampling from a data distribution $p_{\text{data}}(\mathbf{x})$. To that end, one sets the target distribution $p_1(\mathbf{x}) = p_{\text{data}}(\mathbf{x})$, as well as *source* distribution $p_0(\mathbf{x}) = p_{\text{source}}(\mathbf{x})$, such as the standard-normal. To learn to turn a sample from $p_{\text{source}}(\mathbf{x})$ to one from $p_{\text{data}}(\mathbf{x})$, one interpolates noise and data,

$$\mathbf{x}_t = (1 - t) \cdot \mathbf{x}_0 + t \cdot \mathbf{x}_1, \quad \text{where} \quad \mathbf{x}_0 \sim p_0(\mathbf{x}), \ \mathbf{x}_1 \sim p_1(\mathbf{x}) \ \& \ t \sim p_{\text{time}}(t),$$

and passes the mixture and the timestep to a *momentum* neural network that minimizes

$$\mathbb{E}_{\mathbf{x}_0 \sim p_0, \mathbf{x}_1 \sim p_1, t \sim p_{\text{time}}} \left[ \left( v_\theta(\mathbf{x}_t, t) - (\mathbf{x}_1 - \mathbf{x}_0) \right)^2 \right].$$

Once learned, the momentum network can be used to sample from the target distribution by solving the following ordinary differential equation (ODE) with initial conditions,

$$\mathrm{d}\mathbf{x}_t = v_\theta(\mathbf{x}_t, t)\mathrm{d}t, \quad \mathbf{x}_0 \sim p_0(\mathbf{x}),$$

from $t = 0$ to $t = 1$, which results in a sample $\mathbf{x}_1 \sim p_1(\mathbf{x})$. Typically, the ODE is solved numerically, by discretizing the time interval $[0, 1]$ into $N_{\text{step}}$ steps and using, *e.g.,* the Euler method. In this paper, we will use continuous normalizing flows to model the geometry of materials.

---

[1]$\mathcal{S}^*$ denotes the countable union of the products of the set $\mathcal{S}$ with itself. That is, $\mathcal{S}^* = \cup_{n=1}^\infty \mathcal{S}^n$.

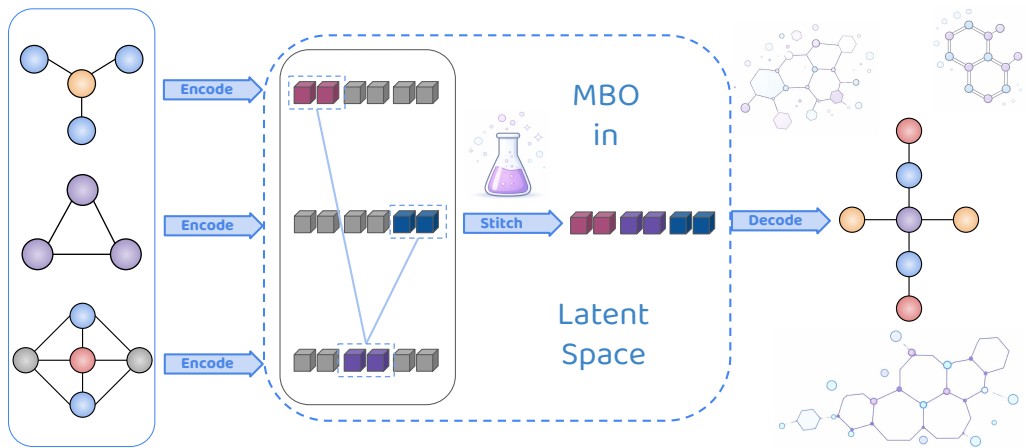

Figure 2: The pipeline of material discovery with MBO in CliqueFlowmer's latent space. Known materials are encoded into a fixed-dimensional latent space that admits a clique decomposition, where each clique contributes additively to the target property. MBO operates by selecting optimal clique instantiations and stitching them together, after which the representation is decoded to generate a new material.

## 2.3 MATERIAL DISCOVERY

We consider a function $f(\mathsf{M}) \in \mathbb{R}$, often reffered to as *target property*, and assume that we are given a finite dataset $\mathcal{D} = \{\mathsf{M}^i, \mathsf{y}^i = f(\mathsf{M}^i)\}_{i=1}^N$ of examples of materials and their values of the property. Given this dataset, our goal is to discover materials that optimize the property,

$$\min_{\mathsf{M} \in \mathcal{M}} f(\mathsf{M}) \qquad (1)$$

without the necessity to evaluate consecutive candidates in a wet lab[2]. That is, we want to do this *fully offline*. The standard approach to this problem consists of two steps: *1)* learning to sample new materials with a generative model $p_\theta(\mathsf{M})$, and *2)* to select the most promosing candidate out of $N_{\mathrm{prop}}$ proposed materials (Chen & Gu, 2020; Park et al., 2024; Zeni et al., 2025). However, this approach is quite inefficient of an optimization algorithm since sampling from $p_\theta(\mathsf{M})$ only explores the regions of the material space that have been seen during the model's training. Thus, new methods that boldly explore attractive regions of the material space are desirable (more background in Appendix A).

In this work, we suggest to solve the optimization problem in Equation (1) directly. That is, we will use tools from offline model-based optimization (MBO) to model the target property $f(\mathsf{M})$ and find candidates for its minimizers. To enable application of these techniques in Material Discovery (MD), for the first time, we introduce a novel neural-network model, dubbed *CliqueFlowmer*, that turns materials into variables tractable by MBO.

## 3 CLIQUEFLOWMER

This section introduces the architecture and the training algorithm of our MBO model for MD. It consists of a few components. In Section 3.1 we introduce an encoder that maps a material M to a continuous latent representation $\mathbf{z}$. Then, Section 3.3 describes a decoder that reconstructs the material—both the atom types and the geometry, from the representation. As we demonstrate later, such a setup allows to parameterize materials as continuous vectors which can be optimized against the target property with gradient-based techniques (see Figure (2)).

### 3.1 ENCODER

Our encoder has to combine four distinct pieces of information: lattice lengths $(a, b, c)$, lattice angles $(\alpha, \beta, \gamma)$, atom positions $\mathbf{X}$, and atom types $\mathbf{a}$ (the latter two being irregular), and produce a fixed-

---

[2]Maximization problems can be represented analogously and solved using the same techniques.

dimensional continuous vector $\mathbf{z} \in \mathbb{R}^{d_{\mathbf{z}}}$. To do it, first, we map the continuous inputs to vectors of size $d_{\text{model}}$ with their corresponding MLPs,

$$\mathbf{h}^{\text{len}} = \text{MLP}_\theta^{\text{len}}(a, b, c), \quad \mathbf{h}^{\text{ang}} = \text{MLP}_\theta^{\text{ang}}(\alpha, \beta, \gamma), \quad \mathbf{H}^{\text{pos}} = \text{MLP}_\theta^{\text{pos}}(\mathbf{X})$$

and we map the atom types to continuous $d_{\text{model}}$-dimensional embeddings $\mathbf{H}^{\text{atom}}$. The MLP-produced hidden states are then concatenated, $\mathbf{H}^{\text{in}} = [\mathbf{h}^{\text{len}}, \mathbf{h}^{\text{ang}}, \mathbf{H}^{\text{pos}}]$, and passed into a transformer, where they are conditioned by $\mathbf{H}^{\text{atom}}$ via adaptive layer-norm (Peebles & Xie, 2023, AdaLN) that replaces layer-norm (Lei Ba et al., 2016) from the standard transformer (Vaswani et al., 2017),

$$\mathbf{H}^{\text{out}} = T_\theta^{\text{enc}}(\mathbf{H}^{\text{in}}, \mathbf{H}^{\text{atom}}).$$

For a single instance of $N_{\text{atom}}$ atoms in the unit cell, the output of the transformer is a tensor of shape $(2 + N_{\text{atom}}) \times d_{\text{model}}$, thus depending on the cell size. To produce a fixed-dimensional representation $\mathbf{z}$, we pool the tensor along the first dimension, using attention with a learnable query vector $\mathbf{q} \in \mathbb{R}^{d_{\text{model}}}$, yielding a hidden state,

$$\mathbf{h}^{\text{pool}} = \text{Att}(\mathbf{q}, \mathbf{K}\mathbf{H}^{\text{out}}, \mathbf{V}\mathbf{H}^{\text{out}}) \in \mathbb{R}^{d_{\text{model}}}, \tag{2}$$

which is further post-processed by GELU (Hendrycks & Gimpel, 2016) and layer-norm, yielding $\mathbf{h}^{\text{post}}$. It is then fed to a linear layer that produces the mean and log-standard deviation parameters of the normal distribution of the latent representation,

$$[\mu_{\mathbf{z}}, \log \sigma_{\mathbf{z}}] = \text{Lin}_\theta(\mathbf{h}^{\text{post}}), \quad \mathbf{z} \sim \mathcal{N}(\mu_{\mathbf{z}}, \sigma_{\mathbf{z}}^2).$$

We highlight that the pooling operation from Equation (2) is what enables us to encode and navigate the transdimensional material space $\mathcal{M}$ in a fixed-dimensional vector space, later enabling employing MBO.

## 3.2 PREDICTOR

Once sampled, in addition to the standard flat form, the latent vector attains a form of a chain of cliques `chain`$(\mathbf{z}, d_{\text{clique}}, d_{\text{knot}})$ of clique size $d_{\text{clique}}$ and knot size $d_{\text{knot}}$. That is, it is a $N_{\text{clique}} \times d_{\text{clique}}$ matrix $\mathbf{Z}$ with entry values from $\mathbf{z}$ such that $\mathbf{Z}_{i,j} = \mathbf{z}_k$, where $k = (i - 1) \cdot (d_{\text{clique}} - d_{\text{knot}}) + j$, *i.e.*, the last $d_{\text{knot}}$ entries of row $i$ and the first $d_{\text{knot}}$ entries of row $i + 1$, for $i = 1, \ldots, N_{\text{clique}} - 1$, are equal. We write $\mathbf{Z}_i$ to denote its $i$th row. The matrix is then fed to an MLP predictor of property $\mathbf{y} = f(\mathsf{M})$ that decomposes its prediction over the cliques of $\mathbf{Z}$,

$$f(\mathsf{M}) \approx f_\theta(\mathbf{z}) = \sum_{c=1}^{N_{\text{cliques}}} f_\theta(\mathbf{Z}_c, c).$$

Imposing such a decomposable structure on the latent space is known to improve the effectieness of MBO (Kuba et al., 2024). Intuitively (see Figure 2), it enables composing optimal in-distribution examples of each clique to form a competitive in-distribution solution—a property also known as *stitching* (Fu et al., 2020; Shin et al., 2025).

## 3.3 DECODER

The decoder consists of two modules: the atom type decoder and the geometry decoder. The former infers the atom types given the latent representation, $p_\theta(\mathbf{a}|\mathbf{z})$, and the latter infers the unit cell geometry given the latent representation and the atom types, $p_\theta(\mathbf{G}|\mathbf{z}, \mathbf{a})$.

**Atom types.** The atom type decoder is quite straightforward—we utilize the standard causal transformer with next-token prediction which we condition on the latent representation via AdaLN. For that end, first, we map the $d_{\mathbf{z}}$-dimensional $\mathbf{z}$ to the $d_{\text{model}}$-dimensional space as

$$\mathbf{z}^{\text{mod}} = \text{LayerNorm}(\text{GELU}(\text{Lin}_\theta(\mathbf{z}))), \tag{3}$$

and then condition with it through AdaLN inside of the causal transformer to produce a hidden state

$$\mathbf{h}_{k+1} = T_\theta^{\text{dec}}(\mathbf{a}_{0:k}, \mathbf{z}^{\text{mod}})_{k+1},$$

for $k = 0, \ldots, N_{\text{clique}}$, where $\mathbf{a}_0$ and $\mathbf{a}_{N_{\text{clique}}+1}$ are $\langle\text{Start}\rangle$ and $\langle\text{Stop}\rangle$ tokens, respectively. That hidden state is then refined into log-likelihoods with an MLP and a log-softmax layer that predicts the next (atom type) token.

**Geometry.** We decode the shape of the unit cell and the atom positions with a continuous normalizing flow, conditioned on $\mathbf{z}$, for which we build another specialized transformer. We use the flow-matching framework, wherein we decode the material geometry $\mathbf{G} = [(a, b, c),\ (\alpha, \beta, \gamma),\ \mathbf{X}]$ by, first, initializing at a sample from a prior distribution

$$\mathbf{G}_0 = [(a, b, c),\ (\alpha, \beta, \gamma),\ \mathbf{X}]_0 \sim p_0(\mathbf{G}|\mathbf{a}),$$

and then solving the ODE

$$\mathrm{d}\mathbf{G}_t = V_\theta(\mathbf{G}_t, t|\mathbf{a}, \mathbf{z})\mathrm{d}t,$$

from $t = 0$ to $t = 1$, where $V_\theta(\mathbf{G}_t, t|\mathbf{a}, \mathbf{z})$ is a transformer denoiser. The prior distribution $p_0(\mathbf{G}|\mathbf{a}) = p_0^{\text{len}}(a, b, c|\mathbf{a})p_0^{\text{ang}}(\alpha, \beta, \gamma|\mathbf{a})p_0^{\text{pos}}(\mathbf{X}|\mathbf{a})$ is modeled meticulously—first, angles are modeled as random uniform in interval $(\pi/3, 2\pi/3)$, and positions are random uniform on $(0, 1)$. The prior over the lengths is chosen so that the density of the unit cell $N_{\text{atom}}/\text{Vol}(\mathsf{M})$ is invariant of the number of atoms, similarly to Zeni et al. (2023). To that end, we observe that $\text{Vol}(\mathsf{M}) \propto abc$ and, thus, compose sampling the lengths from an independent prior, $(\underline{a}, \underline{b}, \underline{c}) \sim \underline{p}^{\text{len}}(\underline{a}, \underline{b}, \underline{c})$, with scaling by $\sqrt[3]{N_{\text{atom}}}$. Lastly, the prior $\underline{p}^{\text{len}}(\underline{a}, \underline{b}, \underline{c})$ over $N_{\text{atom}}$ independent lengths is modeled as a log-normal distribution whose parameters are estimated from training data. For more details, see Appendix D.

Similarly to the encoder, given a noisy unit cell shape and atom positions $\mathbf{G}_t = [(a, b, c), (\alpha, \beta, \gamma), \mathbf{X}]_t$ at time $t$, we embed them in the $d_{\text{model}}$-dimensional space with MLPs. This is then fed to a (non-causal) transformer that is conditioned on the embeddings of the decoded atom types $\mathbf{H}^{\text{atom}}$ via AdaLN. Crucially, each block of the transformer contains a cross-attention layer between the hidden states and the latent representations in form $\mathbf{Z} = \texttt{chain}(\mathbf{z}, d_{\text{clique}}, d_{\text{knot}})$. More specifically, cliques of $\mathbf{Z}$ get first mapped to the $d_{\text{model}}$-dimensional space

$$\mathbf{H}^{\mathbf{z}} = \text{LayerNorm}(\text{GELU}(\text{MLP}_\theta^{\text{lat}}(\mathbf{Z}))),$$

and a cross-attention layer is added between the self-attention and feed-forward layers,

$$\mathbf{H}_t^{\text{cross}} = \mathbf{H}_t^{\text{self}} + \text{CrossAtt}(\mathbf{H}_t^{\text{self}}, \mathbf{H}^{\mathbf{z}}).$$

of the transformer block. The output state $\mathbf{O}_t$ of the transformer is then decomposed into the lengths, angles, and position parts, each of which predicts the corresponding modality with an MLP,

$$\hat{\mathbf{v}}_t^{\text{len}} = \text{MLP}_\theta^{\text{len}}(\mathbf{o}_t^{\text{len}}), \quad \hat{\mathbf{v}}_t^{\text{ang}} = \text{MLP}_\theta^{\text{ang}}(\mathbf{o}_t^{\text{ang}}), \quad \hat{\mathbf{V}}_t^{\text{pos}} = \text{MLP}_\theta^{\text{pos}}(\mathbf{O}_t^{\text{pos}})$$

which together form the prediction from our denoiser, $V_\theta(\mathbf{G}_t, t|\mathbf{a}, \mathbf{z}) = [\hat{\mathbf{v}}_t^{\text{len}}, \hat{\mathbf{v}}_t^{\text{ang}}, \hat{\mathbf{V}}_t^{\text{pos}}]$. Lastly, to sample the denoising timestep for training, we draw a sample from, our novel, *lifted logit-normal* distribution, which is a Bernoulli mixture of the logit-normal and the uniform distributions,

$$t = (1 - \mathsf{b}) \cdot t_{LN} + \mathsf{b} \cdot t_U, \quad \text{where} \quad \mathsf{b} \sim \text{Ber}(\epsilon),\ t_{LN} \sim \text{LogitNorm}(0, 1),\ t_U \sim U[0, 1],$$

to prioritize the challenging denoising tasks from the midpoint timesteps and ensure sufficient coverage of the edge timesteps. We set $\epsilon = 0.1$ (see Figure 3).

### 3.4 TRAINING

The goal of training CliqueFlowmer is to learn a map $p_\theta(\mathbf{z}|\mathbf{a}, \mathbf{G})$ of a material $\mathsf{M} = [\mathbf{a}, \mathbf{G}]$ to a fixed-dimensional continuous vector $\mathbf{z}$, a decomposable approximator $f_\theta(\mathbf{z})$ of the target property, and the inverse $p_\theta(\mathbf{a}, \mathbf{G}|\mathbf{z}) = p_\theta(\mathbf{a}|\mathbf{z})p_\theta(\mathbf{G}|\mathbf{a}, \mathbf{z})$ of the map. Thus, to train CliqueFlowmer, we must combine training of its four integrated submodules. At the nucleus of this fusion there is sampling the latent representation $\mathbf{z} \sim p_\theta(\mathbf{z}|\mathbf{a}, \mathbf{G})$—given this vector, the atom type decoder learns to maximize the conditional next-token log-likelihood

$$-\mathcal{L}_{\text{atom}}(\mathbf{a}, \mathbf{z}) = \sum_{i=0}^{N_{\text{atom}}} \log p_\theta(\mathbf{a}_{i+1}|\mathbf{a}_{0:i}, \mathbf{z}),$$

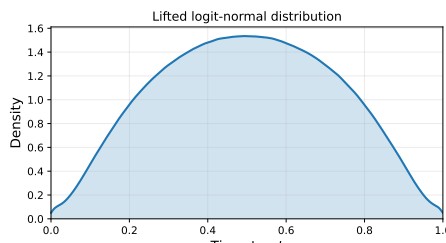

Figure 3: The density plot of the lifted logit-normal distribution. This distribution places more mass on values of $t$ in the middle of the range $(0, 1)$ and puts substantial probability near the endpoints.

where $a_0 = \langle \text{Start} \rangle$ and $a_{N_{\text{atom}}+1} = \langle \text{Stop} \rangle$ are *Start* and *Stop* tokens.

Meanwhile, the flow-matching model that uses cross-attention to conditoin on the clique form $\mathbf{Z}$ of the latent, samples a noise geometry $\mathbf{G}_0$ for a cell with $N_{\text{atom}}$ atoms, interpolates it with the ground-truth geometry at a random timestep, $\mathbf{G}_t = (1-t) \cdot \mathbf{G}_0 + t \cdot \mathbf{G}$, and minimizes the difference between the momentum network's prediction and the path between geometries,

$$\mathcal{L}_{\text{len}}(\mathbf{a}, \mathbf{z}, \mathbf{G}, \mathbf{G}_0) = \left( (a - a_0, b - b_0, c - c_0) - \hat{\mathbf{v}}_t^{\text{len}} \right)^2$$

$$\mathcal{L}_{\text{ang}}(\mathbf{a}, \mathbf{z}, \mathbf{G}, \mathbf{G}_0) = \left( (\alpha - \alpha_0, \beta - \beta_0, \gamma - \gamma_0) - \hat{\mathbf{v}}_t^{\text{ang}} \right)^2$$

$$\mathcal{L}_{\text{pos}}(\mathbf{a}, \mathbf{z}, \mathbf{G}, \mathbf{G}_0) = \sum_{i=1}^{N_{\text{atom}}} \left( (\mathbf{x}^i - \mathbf{x}_0^i) - \hat{\mathbf{v}}_t^{\text{pos},i} \right)^2.$$

While training the flow denoiser, we mask the latent $\mathbf{z}$ and replace it with a random normal noise, $\epsilon_{\mathbf{z}} \sim \mathcal{N}(\mathbf{0}_{d_{\mathbf{z}}}, \mathbf{I}_{d_{\mathbf{z}}})$ with probability $p_{\text{lat}} = 0.1$. We decompose the flow loss into distinct components to ensure that the unit cell (lengths and angles) receives equal weight for every material in the batch. For clarity, we define the sum of flow losses as

$$\mathcal{L}_{\text{flow}}(\mathbf{a}, \mathbf{z}, \mathbf{G}, \mathbf{G}_0) = \mathcal{L}_{\text{len}}(\mathbf{a}, \mathbf{z}, \mathbf{G}, \mathbf{G}_0) + \mathcal{L}_{\text{ang}}(\mathbf{a}, \mathbf{z}, \mathbf{G}, \mathbf{G}_0) + \tau_{\text{pos}} \cdot \mathcal{L}_{\text{pos}}(\mathbf{a}, \mathbf{z}, \mathbf{G}, \mathbf{G}_0),$$

where $\tau_{\text{pos}}$ is a positive scalar. Further, the latent represtantation, in its clique form, is also tasked with minimizing the prediction error of the target property,

$$\mathcal{L}_{\text{pred}}(\mathsf{M}, \mathbf{z}) = \left( f_\theta(\mathbf{z}) - f(\mathsf{M}) \right)^2,$$

and the clique-based KL-divergence is computed for the parameters of the normal distribution of $\mathbf{z}$,

$$\mathcal{L}_{\text{lat}}(\mathsf{M}, \mu_{\mathbf{z}}, \sigma_{\mathbf{z}}) = \text{KL}\left( \mathcal{N}(\mu_{\mathbf{z},c}, \sigma_{\mathbf{z},c}^2), \, \mathcal{N}(\mathbf{0}_{d_{\text{clique}}}, \mathbf{I}_{d_{\text{clique}}}) \right), \tag{4}$$

where the clique $c$ is drawn from the uniform distribution $U[N_{\text{clique}}]$ (Kuba et al., 2024). Summarizing, and writing $\mathsf{M} = [\mathbf{a}, \mathbf{G}]$ where possible, the total expected loss is

$$\mathcal{L}(\theta) = \mathbb{E}_{\mathsf{M} \sim \mathcal{D}, \mathbf{z} \sim p_\theta, \mathbf{G}_0 \sim p_0} \left[ \mathcal{L}_{\text{atom}}(\mathbf{a}, \mathbf{z}) + \mathcal{L}_{\text{flow}}(\mathsf{M}, \mathbf{z}, \mathbf{G}_0) + \tau_{\text{pred}} \mathcal{L}_{\text{pred}}(\mathsf{M}, \mathbf{z}) + \beta \mathcal{L}_{\text{lat}}(\mathsf{M}, \mu_{\mathbf{z}}, \sigma_{\mathbf{z}}) \right],$$

where $\beta$ and $\tau_{\text{pred}}$ are positive scalars which we warm up lineary one after another.

## 3.5 MATERIAL DISCOVERY WITH CLIQUEFLOWMER

To optimize new materials with our model, we conduct a form of gradient-based search in the representation space. We initialize this process by encoding a sample of existing materials with the encoder, $\mathbf{z} = \text{Enc}_\theta(\mathbf{a}, \mathbf{G})$. Then, we optimize the prediction of the target property,

$$\mathbf{z}_\star = \arg\min_{\mathbf{z}^{\text{new}}} f_\theta(\mathbf{z}^{\text{new}}). \tag{5}$$

The choice of the minimization algorithm plays a significant role. While we have found that exact back-propagation of Equation (5) is prone to adversarial exploitation of the model, back-propagation-free evolution strategies (Salimans et al., 2017, ES) was very effective (see Section 4.2). That is, at every iteration, we draw $N_{\text{pert}}$ noise vectors $\epsilon$ from the standard normal distribution, $\epsilon \sim \mathcal{N}(\mathbf{0}_{d_{\mathbf{z}}}, \mathbf{I}_{d_{\mathbf{z}}})$, and compute the predicted values $f_\theta(\mathbf{z} + \sigma \epsilon)$ of the perturbations of $\mathbf{z}$ at scale $\sigma$. Then, we rank these perturbations, $\epsilon^1, \ldots, \epsilon^{N_{\text{pert}}}$, based on the predicted values of $f$ (so that the smallest value gets the lowest rank). We standardize the ranks to have zero mean and unit variance, $i \mapsto R^i$, for $i = 1, \ldots, N_{\text{pert}}$, and compute the ES gradient as

$$\hat{\nabla}^{\text{ES}}(\mathbf{z}) = \frac{1}{N_{\text{pert}} \sigma} \sum_{i=1}^{N_{\text{pert}}} R^i \epsilon^i. \tag{6}$$

Additionally, we use antithetic sampling (see Appendix B for more information about ES). Then, we take a gradient step with AdamW optimizer (Loshchilov et al., 2017). The weight decay step of the optimizer, $\mathbf{z} \mapsto (1 - \lambda)\mathbf{z}$, is crucial since it brings the optimized latent variable closer to the origin. In effect, this step increases the likelihood of the latent under the Gaussian prior that CliqueFlowmer was trained with (see Equation (4)), which mitigates the problem of going out of distribution with $\mathbf{z}$.

| Metric | CrystalFormer | DiffCSP | MatterGen | CliqueFlowmer | CliqueFlowmer-Top |
|--------|---------------|---------|-----------|---------------|-------------------|
| Eform ($\downarrow$) | 0.71 | 0.62 | 0.60 | -1.07 | **-1.19** |
| S.U.N. ($\uparrow$) | 12.8 | 18.4 | 15.7 | 0.4 | 0.0 |
| Band Gap ($\downarrow$) | 0.52 | 0.59 | 0.46 | 0.07 | **0.05** |
| S.U.N. ($\uparrow$) | 12.8 | 18.4 | 15.7 | **48.9** | 43.9 |

Table 1: Target property values (Eform and Band Gap) and S.U.N. rate (percentage) across generative model baselines and MBO with our model. CliqueFlowmer and CliqueFlowmer-Top reduce formation energy from the initial sample average of 0.46 to -1.07 and -1.19, respectively. Notably, however, the produced materials achieve very low S.U.N. rates. CliqueFlowmer and CliqueFlowmer-Top also reduce the band gap from the sample average of 0.57 to 0.07 and 0.05, respectively, while achieving superior S.U.N. rates as well. Thus, our models solve the optimization problem in Equation (1) but the stability of proposed materials depends on the optimization objective.

Given an optimized latent representation $\mathbf{z}_\star$, we modulate it with Equation (3), and pass to the atom-type decoder, from which we sample the atom sequence with beam search of width $N_{\text{beam}} = 10$,

$$\mathbf{a}_\star \sim \text{BeamSearch}(T_\theta^{\text{dec}}, \mathbf{z}_\star^{\text{mod}}).$$

Next, we consider the clique form of the latent representation, $\mathbf{Z}_\star = \texttt{chain}(\mathbf{z}_\star, d_{\text{clique}}, d_{\text{knot}})$, and pass it to our continuous normalizing flow model, from which we sample the material geometry. For this end, we use the *Euler method* with *classifier-free guidance* (Ho & Salimans, 2022, CFG), in which the effective momentum used in the ODE is

$$V_\theta^\omega(\mathbf{G}_t, t, \mathbf{z}_\star) = (1 + \omega) \cdot V_\theta(\mathbf{G}_t, t, \mathbf{z}_\star) - \omega \cdot V_\theta(\mathbf{G}_t, t, \epsilon_{\mathbf{z}}),$$

where $\epsilon_{\mathbf{z}}$ is standard normal noise. In our experiments, we used $\omega = 2$ (ablations in Appendix C).

## 4 EXPERIMENTS

We tested CliqueFlowmer in two tasks for which we trained two separate models. Both of them were trained on the MP-20 dataset that contains 45K example materials in total (Xie et al., 2021). In the first task, for the role of the target property, we chose the predicted formation energy per atom from M3GNet (Chen & Ong, 2022),

$$f(\mathsf{M}) = \frac{\mathcal{E}_{\text{form}}(\mathsf{M})}{N_{\text{atom}}},$$

and for the second one, the band gap predicted by MEGNet (Chen et al., 2019),

$$f(\mathsf{M}) = \Delta_{\text{band}}(\mathsf{M}).$$

It is worth noting that, for material scientists, formation energy is not the most useful property to minimize since it doesn't translate to any extravagant physical behavior. Nevertheless, formation energy oracles, such as M3GNet, are easily accessible and accurate (in fact more accurate than band gap oracles), and thus suffice to demonstrate the capability of our method. In the following subsections, we evaluate the CliqueFlowmer's abilities in material discovery.

### 4.1 MBO WITH CLIQUEFLOWMER

To test CliqueFlowmer's ability to conduct MD through MBO, we use the following protocol. We sample $N$ existing materials and encode them with the CliqueFlowmer encoder. Then, we optimize the materials' representations, using $T = 2000$ gradient steps (Equation (6)). The converged representations are then decoded with the CliqueFlowmer decoder, thus rendering $N$ new materials (see Figure (5) for examples). Additionally, one can filter out the most promising candidates prior to their ground-truth evaluation. Namely, once optimization converges, one can predict the target property with the prediction head, $f_\theta(\mathbf{z})$, and select top-$k\%$ solutions with the best values. The attractiveness of this approach stems from the computational savings it offers—having run the cheap optimization stage in the latent space, one can reject the majority of suboptimal latents before passing the most promising ones to the expensive denoising and relaxation stages. We refer to this approach *CliqueFlowmer-Top* and set $k = 10$ in our experiments.

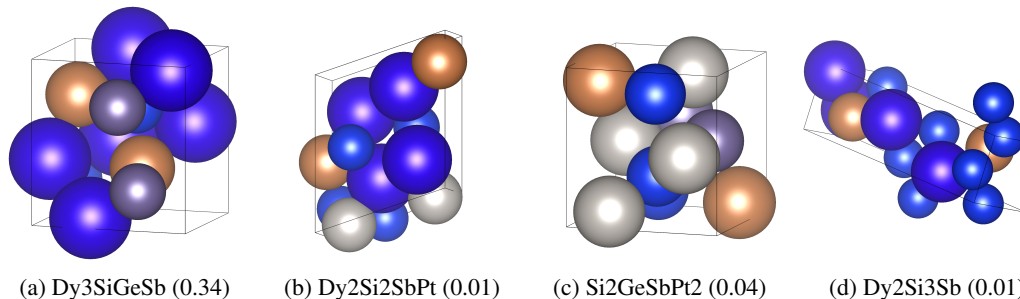

| (a) Dy3SiGeSb (0.34) | (b) Dy2Si2SbPt (0.01) | (c) Si2GeSbPt2 (0.04) | (d) Dy2Si3Sb (0.01) |

Figure 5: Some of the materials optimized by CliqueFlowmer for band gap minimization. Each figure shows a material's unit cell, and the caption describes its composition and band gap values in format `composition(band gap)`. We can observe that the optimized materials often include Dysprosium (Dy) and Silicon (Si) as components.

To test the success of this optimization procedure, we measure the target property value of the $N = 10^4$ original materials and the value of the newly-proposed ones. Additionally, to verify that our method produces useful materials, we calculate the S.U.N. rate (stable, unique, novel) (Zeni et al., 2023) among the proposed solutions. We compare our results to those of CrystalFormer (Taniai et al., 2024), DiffCSP (Jiao et al., 2023), and MatterGen (Zeni et al., 2023), whose generated structures we obtained from Kazeev et al. (2025). Due to computational limitations, we relax the materials approximately with M3GNet (Chen & Ong, 2022) and determine stability with energy above local hull. Results in Table 1 show that CliqueFlowmer drastically reduces the value of the target property (M3GNet formation energy

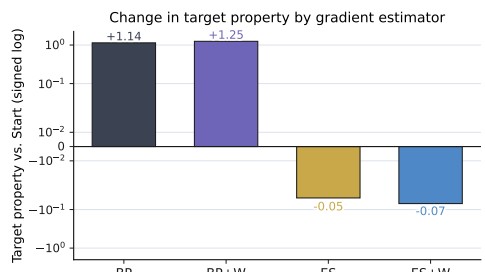

Figure 4: Comparison of gradient estimators—back-propagation (BP) vs. evolution strategies (ES), and weight decay, against the target property. We plot the average change of the target property, over 100 materials, in log-scale. Each algorithm performed 1000 steps, which was sufficient to reject back-propagation as divergent.

& MEGNet band gap), significantly outperforming materials generated by the baselines. This quality is further amplified by CliqueFlowmer-Top, which reduces the objective value even further. Furthermore, while the purpose of CliqueFlowmer was property optimization, the structures proposed by the model outperform the baselines in terms of the S.U.N. rate in the band gap optimization task and underperform in the formation energy task. We attribute this result to the implications of the optimization objective on stability—simply minimizing the formation energy may lead to low-energy regions of the material space but not necessarily to the lowest-energy material[3]. Nevertheless, the strong optimization performance and the very low S.U.N. rate of band gap materials exceeded our expectations. We attribute this result to our careful construction of our domain-specific auto-encoder and conservative latent-space optimization (see Appendix E & G for more information).

### 4.2 OPTIMIZATION ALGORITHM

As described in Section 3.5, having trained CliqueFlowmer, we discover new materials by solving the optimization problem from Equation (5). In this paper, we utilized the back-propagation-free ES algorithm (Salimans et al., 2017). This section provides an empirical justification of this counterintuitive choice. We compare the performance of the gradient derived by back-propagation (BP) to ES by optimizing $N = 100$ structures, in CliqueFlowmer's latent space, for $T = 1000$ steps. In our experiments, both gradients were applied by Adam optimization algorithm (Kingma, 2014), as well as by its decoupled weight decay variant (Loshchilov et al., 2017, BP+W, ES+W). Figure (4) shows that, instead of decreasing the minimized property, BP and BP+W have drastically increased it. Meanwhile, ES has successfully progressed towards the property minimization, and ES+W has done that most effectively. Thus, our main experiments use ES+W.

---

[3] A material's stability is determined by energy above hull where materials of a given composition that attain lowest energy tend to be more stable.

Consult Appendix E for more experiments on the optimization stage, Appendix F for our analysis of the learned representations, and Appendix G for recommended hyper-parameters.

## 5 RELATED WORKS

The cost and duration of material discovery (MD) has motivated research in automated methods (Jain et al., 2013). Recently, thanks to advancements in generative models, deep learning-based techniques of direct discovery through generation have become popular. Notably, Crystal Diffusion VAE (Xie et al., 2021, CDVAE) leverages diffusion models (Ho et al., 2020) to harness sampling of material representations learned by a variational auto-encoder (Kingma & Welling, 2013) which, unlike ours, are not equipped with a decomposable structure. Similarly, FlowMM (Miller et al., 2024) utilizes flow-matching (Lipman et al., 2022) models to directly generate new materials. While we also do use flows, that work's main focus is mathematically sound generative modeling on appropriate manifolds that materials are embedded in, while we use flows as decoders of materials optimized in a latent space. Additionally, both of these models utilize the expensive equivariant graph neural networks which, unlike small transformers, do not fit into the computational budget of many researchers (Li et al., 2025). Further, CrystalFormer (Taniai et al., 2024) generates materials auto-regressively, atom by atom. We generate atom species autoregressively in our model, but the geometry is generated with a continuous normalizing flow. Most importantly, in our MBO model, these components are not the core generators, but they are just decoders of latent representations of optimized materials. MatterGen (Zeni et al., 2023), similarly to CDVAE, generates materials with diffusion models, and enables doing so conditioned on target properties. CliqueFlowmer, instead, directly optimizes the property value. Similarly to us, All-atom Diffusion Transformers (Joshi et al., 2025, ADiT) combine transformers and latent variables. In their architecture, however, the dimensionality of the latent variables depends on the atom count of a material at hand. This, in addition to not offering compression abilities, prevents employing MBO in the transcdimensional material space. Meanwhile, in CliqueFlowmer, all materials get compressed to a continuous, fixed-dimensional latent vector which enables gradient-based search of the material space.

There is a growing line of work in molecule discovery with diffusion models where the samplers are steered towards a target property-based Boltzmann distribution (Li et al.; Uehara et al., 2024; Tan et al., 2025; Liu et al., 2025). While these methods enable tilting the distribution of the target property to the desired side, we propose a method to directly optimize it in the space of chemical materials. Notably, MBO methods, such as MINs (Kumar & Levine, 2020), COMs (Trabucco et al., 2021), or MatchOpt (Hoang et al., 2025), aim at directly optimizing target property values. Among such methods, Cliqueformer (Kuba et al., 2024)—having introduced clique-based MBO into transformers—is most similar to our work. However, neither Cliqueformer nor prior MBO methods are compatible with material discovery (MD) due to their simple neural network architectures suitable only for MBO benchmarks. Our work transcends these limitations by building a domain-specific model and learning algorithm that enables direct property optimization in MD.

## 6 CONCLUSION

In this work, we proposed addressing the problem of material discovery (MD) with the tools from offline model-based optimization (MBO). We introduced CliqueFlowmer—a model that represents materials as continuous vectors and optimizes them with clique-based MBO. Our experimental results with MD aparatus, such as MP-20 data and target property oracles, show that CliqueFlowmer enables efficient optimization of materials offline. The major limitation of our work is the impact of optimized properties—while the most easily measurable with oracles, they are not of the greatest interest of industrial labs that design materials for specialized applications. Together with increased access to oracles of more properties, we expect CliqueFlowmer to solve more impactful problems. Another limitation is the accuracy of our evaluation. For example, some works estimate properties of generated materials with software conducting (high-cost) density functional theory calculations which are known to be very accurate, while we employed machine learning-based oracles which can induce errors. Nevertheless, our experiments are rigorous instances of MBO tasks in the material space, and thus confirm the validity of our method. Lastly, we hope that this work will inspire the MD community to add MBO to its toolkit.

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

# A ADDITIONAL BACKGROUND

## A.1 MATERIAL DISCOVERY PRACTICES

Material discovery (MD) aims to identify atomic structures whose physical or chemical properties optimize a desired objective, such as formation energy, stability, or electronic behavior. In inorganic materials science, materials are typically represented as periodic crystal structures with a unit cell geometry and a variable number of atoms with associated species and positions. Evaluating candidate materials using first-principles methods such as density functional theory (DFT) is accurate but computationally expensive, motivating the construction of large offline datasets such as the Materials Project (Jain et al., 2013).

Recent machine-learning approaches to MD broadly fall into two categories. *Surrogate modeling* methods learn predictors of material properties and use them for screening or ranking candidates. *Generative modeling* methods—based on VAEs, diffusion models, or flows—learn to sample new materials resembling those in a reference dataset (Xie et al., 2021; Zeni et al., 2023; Taniai et al., 2024). While generative models can produce valid and diverse structures, their likelihood-based training objective concentrates probability mass near the empirical data distribution, limiting their ability to aggressively explore property-optimal regions.

Offline model-based optimization (MBO) offers an alternative paradigm by directly optimizing a learned surrogate objective using only offline data (Kumar et al., 2020; Trabucco et al., 2022). Applying MBO to MD is challenging due to the hybrid discrete–continuous structure of materials and their transdimensionality. CliqueFlowmer addresses this challenge by learning a fixed-dimensional latent representation of materials that admits structured optimization and can be decoded back into valid crystal structures.

## A.2 NEXT-TOKEN PREDICTION WITH TRANSFORMERS

Next-token prediction is a standard training paradigm for autoregressive sequence models, including transformers (Vaswani et al., 2017). Given a sequence of discrete tokens $(x_1, \ldots, x_T)$, a causal transformer models the factorized distribution

$$p(\mathbf{x}_1, \ldots, \mathbf{x}_T) = \prod_{t=1}^{T} p(\mathbf{x}_t \mid \mathbf{x}_{<t}),$$

and is trained by minimizing the negative log-likelihood of each token conditioned on its prefix.

In CliqueFlowmer, next-token prediction is used to model the sequence of atom types in a material. Each structure is represented as a variable-length sequence augmented with explicit $\langle\text{Start}\rangle$ and $\langle\text{Stop}\rangle$ tokens. The autoregressive transformer is conditioned on the latent representation via adaptive layer normalization (AdaLN) (Peebles & Xie, 2023), allowing global structural information to influence all token predictions. This approach provides a flexible mechanism for handling discrete, variable-length outputs while integrating naturally with transformer architectures and beam search at generation time.

## A.3 FUNCTIONAL GRAPHICAL MODELS AND CLIQUE-BASED REPRESENTATIONS

CliqueFlowmer builds on the framework of *functional graphical models* introduced by Grudzien et al. (2024). In this framework, the target function of interest is modeled as an additive decomposition over overlapping cliques of latent variables. Concretely, a latent vector $\mathbf{z} \in \mathbb{R}^{d_z}$ is reshaped into a chain of overlapping cliques

$$\mathbf{Z} = \text{chain}(\mathbf{z}, d_{\text{clique}}, d_{\text{knot}})$$

where each clique shares a subset of variables (knots) with its neighbors. The surrogate objective is then parameterized as

$$f_\theta(\mathbf{z}) = \sum_{c=1}^{N_{\text{cliques}}} f_\theta(\mathbf{Z}_c, c),$$

where each term depends only on a local clique.

This structured decomposition induces a functional graphical model in which each clique corresponds to a factor, enabling *compositional generalization*. In particular, clique-based models support *stitching*: optimal in-distribution configurations of individual cliques can be recombined to form globally competitive solutions (Fu et al., 2020; Kuba et al., 2024). This property has been shown to substantially improve the effectiveness of offline MBO in high-dimensional design spaces.

In CliqueFlowmer, this clique structure is imposed on the latent representation of materials, enabling gradient-based or derivative-free optimization directly in latent space while maintaining a strong inductive bias toward in-distribution solutions. The decoder then maps optimized latent variables back into discrete atom types and continuous geometries, allowing clique-based MBO to operate over the transdimensional space of materials.

## B  EVOLUTION STRATEGIES

Evolution Strategies (ES) are a class of black-box optimization methods that estimate gradients of an expected objective using random perturbations of the parameters rather than back-propagation through the objective itself. In the scalable formulation introduced by Salimans et al. (2017), ES optimizes parameters $\phi \in \mathbb{R}^d$ by minimizing the smoothed objective

$$F(\phi) = \mathbb{E}_{\epsilon \sim \mathcal{N}(0,I)}\big[f(\phi + \sigma\epsilon)\big], \tag{7}$$

where $\sigma$ controls the perturbation scale. The gradient of $F(\phi)$ can be estimated via Monte Carlo sampling thanks to the following identity,

$$\nabla_\phi F(\phi) = \frac{1}{\sigma}\mathbb{E}_{\epsilon \sim \mathcal{N}(0,I)}\big[f(\phi + \sigma\epsilon)\epsilon\big]$$

and used in a standard stochastic gradient descent loop. In our work, we substitute a representation $\mathbf{z}$ of a material M into the place of $\phi$, and we minimize the expected perturbed approximation $f_\theta(\mathbf{z})$ of the target property.

**Rank-Based ES.**  A key practical component of modern ES is *rank-based fitness shaping*. Instead of using raw function values $f(\phi + \sigma\epsilon_i)$, ES replaces them with normalized ranks computed across the population of perturbations. For example, if the values of, say 3, perturbed parameters ($\phi + \sigma\epsilon_1, \phi + \sigma\epsilon_2, \phi + \sigma\epsilon_3$) are $(10, -1, 5)$, then the values are turned into ranks $(3, 1, 2)$. Then, they are standardized to have mean zero and unit variance, ultimately taking on values $(-\sqrt{1.5}, 0, \sqrt{1.5})$. This transformation makes the update invariant to monotone rescalings of the objective, improves robustness to outliers and heavy-tailed noise, and stabilizes optimization when the scale of $f$ varies over time (Salimans et al., 2017; Wierstra et al., 2014). As a result, ES behaves less like a noisy gradient estimator and more like a robust search-direction method.

**Antithetic Sampling.**  To further reduce estimator variance, ES commonly employs *antithetic sampling*. For each sampled perturbation $\epsilon_i$, its negation $-\epsilon_i$ is also evaluated, producing paired function values $f(\phi + \sigma\epsilon_i)$ and $f(\phi - \sigma\epsilon_i)$. The gradient estimate aggregates these symmetric evaluations. In the function value case, this causes odd-order noise terms to cancel in expectation and significantly reducing variance. When used with ranks, it verifies for each perturbation whether to go along or against the direction it points to.

For completeness, we write down the rank-based ES estimator with antithetic sampling for $N_{\text{pert}}$ perturbations. First, note that each perturbation $\epsilon$ produces values $f(\phi + \sigma\epsilon)$ and $f(\phi - \sigma\epsilon)$, resulting in $2N_{\text{pert}}$ values from all perturbations. We rank all these values in the ascending order (the lowest value gets the lowest rank). Then, for clarity, we rank the perturbations $\epsilon$ based off the ranks of $f(\phi + \sigma\epsilon)$ (where the perturbation) was added, giving $\epsilon^1, \ldots, \epsilon^{N_{\text{pert}}}$. Let $r(i, +)$ and $r(i, -)$ be the ranks of $f(\phi + \sigma\epsilon^i)$ and $f(\phi - \sigma\epsilon^i)$ among all values. Then, let $R_+^i$ and $R_-^i$ be their standardized versions. The final estimator takes form

$$\hat{\nabla}^{\text{ES}}(\phi) = \frac{1}{2\sigma \cdot N_{\text{pert}}} \sum_{i=1}^{N_{\text{pert}}} \big(R_+^i - R_-^i\big)\epsilon^i.$$

Antithetic sampling effectively doubles sample efficiency and is especially beneficial in high-dimensional settings (Salimans et al., 2017; Glasserman, 2004).

Together, rank-based fitness shaping and antithetic sampling enable ES to scale to millions of parameters while remaining simple, highly parallelizable, and entirely gradient-free, making it a compelling alternative when back-propagation is unavailable, unreliable, or misleading (like in our case).

## C  CLASSIFIER-FREE GUIDANCE FOR DIFFUSION AND FLOW MODELS

Classifier-free guidance (CFG) is a technique originally introduced in the context of diffusion models to control the trade-off between sample quality and diversity without requiring an explicit classifier (Ho & Salimans, 2022). The key idea is to train a single conditional generative model that can operate both conditionally and unconditionally, and to combine the two modes at sampling time to bias generation toward the conditioning signal.

Formally, let $V_\theta(\mathbf{x}_t, t \mid c)$ denote a diffusion or flow model conditioned on some context $c$, and let $V_\theta(\mathbf{x}_t, t \mid \varnothing)$ denote the same model evaluated without conditioning. Classifier-free guidance constructs a guided vector field

$$V_\theta^\omega(\mathbf{x}_t, t \mid c) = (1 + \omega)\, V_\theta(\mathbf{x}_t, t \mid c) \; - \; \omega\, V_\theta(\mathbf{x}_t, t \mid \varnothing),$$

where $\omega \geq 0$ is the guidance strength. Increasing $\omega$ amplifies features correlated with the conditioning signal, at the cost of reduced diversity and potential distributional shift. Of course, $\omega = 0$ is equivalent to not using CFG—it corresponds to the vanilla Euler method.

While CFG was originally developed for diffusion models, the same principle applies directly to continuous normalizing flows and flow-matching models, where the learned vector field defines an ordinary differential equation (ODE) rather than a stochastic reverse process (Lipman et al., 2022). In this setting, guidance modifies the deterministic velocity field used during integration, biasing trajectories toward regions favored by the conditioning signal.

**CFG in CliqueFlowmer.** In CliqueFlowmer, classifier-free guidance is used during decoding of material geometry with the flow-matching model. The conditioning signal is the optimized latent representation $z$, while the unconditional model is obtained by replacing $z$ with Gaussian noise $\varepsilon_z \sim \mathcal{N}(0, I)$. During sampling, the guided vector field is integrated from $t = 0$ to $t = 1$ using an explicit ODE solver.

**Reconstruction Experiment.** To study the effect of guidance strength, we conducted an encode–decode consistency experiment. Given a material M, we encoded it into a latent representation $\mathbf{z}$ using the CliqueFlowmer encoder and then decoded it back into a material $\hat{\mathsf{M}}$ using the geometry flow with different guidance strengths $\omega \in \{0, 2, 4\}$. We then evaluated whether M and $\hat{\mathsf{M}}$ represent the same crystal structure using `StructureMatcher` from `pymatgen`, which accounts for lattice symmetries, atomic species, and fractional coordinates.

We report the *match ratio*, defined as the fraction of decoded structures that were deemed equivalent to their original inputs, over the sample of 100 structures. Results are shown in Table 2.

Table 2: Effect of classifier-free guidance strength on encode–decode consistency.

| Guidance strength $\omega$ | Match ratio (%) |
| --- | --- |
| 0 | 73 |
| 2 | **83** |
| 4 | 77 |

**Discussion.** Moderate classifier-free guidance ($\omega = 2$) substantially improves reconstruction fidelity compared to no guidance, indicating that conditioning on the latent representation is underutilized without explicit amplification. However, excessive guidance ($\omega = 4$) degrades performance, likely due to over-sharpening of the vector field and reduced robustness to modeling errors. This behavior mirrors observations in diffusion-based image and molecule generation, where intermediate

guidance strengths often yield the best balance between faithfulness and stability (Ho & Salimans, 2022).

Based on these results, all main experiments in this work use a guidance strength of $\omega = 2$ during geometry decoding.

**Note.** It was extremely difficult to construct this network architecture and its training regime. The progress towards 80% reconstruction rate was very slow—starting from 20%, and being stuck at 60% for a long time. The usage of CFG is, in fact, one of the detailed factors that can be easily presented in the paper.

## D  PRIOR DISTRIBUTION OF LATTICE LENGTHS

When we decode the geometry $\mathbf{G}$ of a material represented by $\mathbf{z}$ with composition $\mathbf{a}$, we initialize it with a sample from a prior distribution

$$p_0(\mathbf{G}|\mathbf{a}) = p_0^{\text{len}}(a, b, c|\mathbf{a}) \cdot p_0^{\text{ang}}(\alpha, \beta, \gamma|\mathbf{a}) \cdot p_0^{\text{pos}}(\mathbf{X}|\mathbf{a}). \tag{8}$$

Similarly to Miller et al. (2024), we chose to model $p_0^{\text{len}}(a, b, c|\mathbf{a}) = p_0^a(a|\mathbf{a}) \cdot p_0^b(b|\mathbf{a}) \cdot p_0^c(c|\mathbf{a})$ as an independent log-normal distribution. That is, for every $l \in \{a, b, c\}$, we consider it transformation of a variable $l_{\text{nor}}$ via the logarithmic function,

$$l = \exp(l_{\text{nor}}) \quad \text{equiv.} \quad l_{\text{nor}} = \log(l) \tag{9}$$

and we model $l_{\text{nor}}$ as normally distributed, $l_{\text{nor}} \sim \mathcal{N}(\mu_1, \sigma_l^2)$.

That said, we introduce a small modification that accounts for the physical meaning of this distribution. First, we recall that the volume of material M's unit cell is

$$\text{Vol}(\mathsf{M}) = abc \cdot \sqrt{1 - \cos^2 \alpha - \cos^2 \beta - \cos^2 \gamma + 2 \cos \alpha \cos \beta \cos \gamma} \; \propto \; abc. \tag{10}$$

The distribution we described in Equations (8) & (9) does not account for the number of atoms that the initialized unit cell is meant to fit. Meanwhile, materials with more atoms clearly need larger, in terms of volume, cells. To impose it, we introduce a simple change—we modify our prior distribution so that the average atom density,

$$\text{Den}(\mathsf{M}) = \frac{N_{\text{atom}}}{\text{Vol}(\mathsf{M})} \propto \frac{N_{\text{atom}}}{abc}. \tag{11}$$

is invariant to the atom count. We accomplish that by constructing *canonical* lattice lengths, $(\underline{a}, \underline{b}, \underline{c})$, such that $\text{Vol}(\mathsf{M}) \propto N_{\text{atom}}\underline{abc}$, and thus

$$\text{Den}(\mathsf{M}) = \frac{N_{\text{atom}}}{\text{Vol}(\mathsf{M})} \propto \frac{N_{\text{atom}}}{N_{\text{atom}}\underline{abc}} = \frac{1}{\underline{abc}}. \tag{12}$$

To construct them, simply, for each variable $l \in \{a, b, c\}$, we define

$$\underline{l} = \frac{1}{\sqrt[3]{N_{\text{atom}}}} \cdot l \quad \text{equiv.} \quad l = \sqrt[3]{N_{\text{atom}}} \cdot \underline{l},$$

which immediately gives us Equation (12). From then on, we model the canonical lengths as log-normal and esimate their distribution parameters (normal mean and standard deviation from data).

We investigated if this prior fits the data better. Namely, we estimated the distribution of the lattice lengths in two ways—*(1)* without the canonical (atom count-aware) transformation, and *(2)* with the canonical transformation. In (1), the lengths were modeled as log-normal, and in (2), they were modeled as atom count-scaled log-normal variables. We demonstrate the improved quality of method (2) in Figure (6).

## E  OPTIMIZATION ALGORITHM TUNING

We verified, in Section 4.2 (Figure (4), that latent-space optimization of materials with backpropagation-based (BP) gradients diverges and gradients based on rank-based evolution strategies (Salimans et al., 2017, ES) are effective. In this section, we investigate this phenomenon deeper

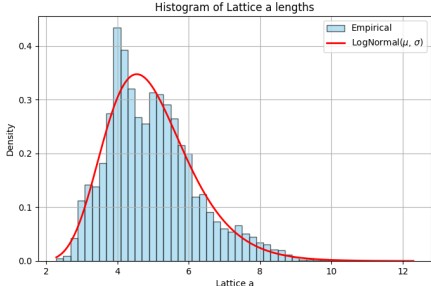 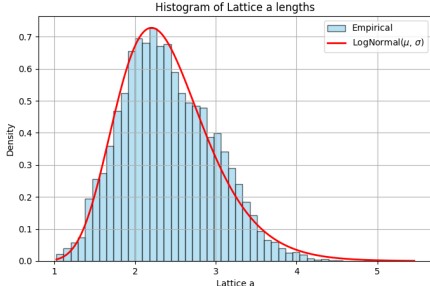

Figure 6: Two ways to model the distribution of lattice lengths, demonstrated on variable $a$. In the first scheme (left), the length is naively modeled as log-normally distributed. In the second, final scheme (right), the length is scaled down by the cubic root of the atom count, and the canonical length is modeled as a logit-normal variable. The log-normal density (red) fits the histogram (blue) better in the canonical case (right).

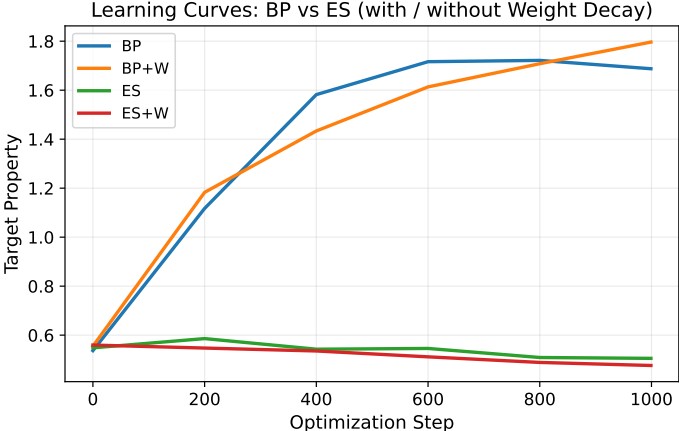

Figure 7: Comparison of gradient estimation algorithms: back-propagation (BP) vs. evolution strategies (ES), with or without weight decay $\lambda = 0.1$. ES stably converges while BP completely diverges.

and ablate the weight decay strength of Adam optimization that we use (Kingma, 2014; Loshchilov et al., 2017). First, we discuss the failure of back-propagation (BP) from Section 4.2 which we additionally illustrate by plotting the learning curves from that experiment in Figure (7). It is known that MBO algorithms are sensitive to distribution shift, wherein optimized designs $\mathbf{x}$ become inputs to a predictive model $f_\theta(\mathbf{x})$ that makes erroneous predictions about them (Kumar & Levine, 2020; Kumar et al., 2020; Trabucco et al., 2021). Meanwhile, back-propagation is the most exact way of calculating the steepest direction of change with respect to such an imprecise function approximator. As a result, a gradient-based algorithm that fully "trusts" this estimator, commits to the artifacts of its errors, leading to poor designs (see Figure (8) for intuition).

We evaluated weight decay values ranging in $\lambda \in \{0, 0.1, 0.2, 0.3, 0.4, 0.5\}$ by applying them to ES and AdamW in a process of optimization of $N = 100$ random materials (without relaxation). This time, we stress-tested the algorithms and ran $T = 2000$ optimization steps (twice as many as in Section 4.2). As a result, we found weights $\lambda = \{0, 0.1\}$ to be less robust than in Section 4.2. On the contrary, we found higher values of $\lambda$ very effective. The gains from increasing the decay weight, however, plateau after $\lambda = 0.3$. Nevertheless, we found $\lambda = 0.5$ to perform best, and thus use it in our experiments.

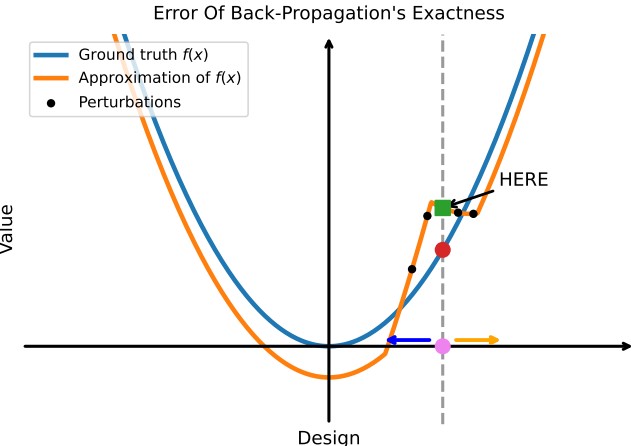

Figure 8: Inuition behind performance discrepancy between BP and ES. The function approximator (orange) of the ground truth function (blue) has an artifact that is inconsistent with the ground truth. Near that artifact's location (pink point), the derivative of the approximator (taken with respect to the green square) points updating to the right. Meanwhile, exploring the region with ES allows to identify the correct (left) update direction.

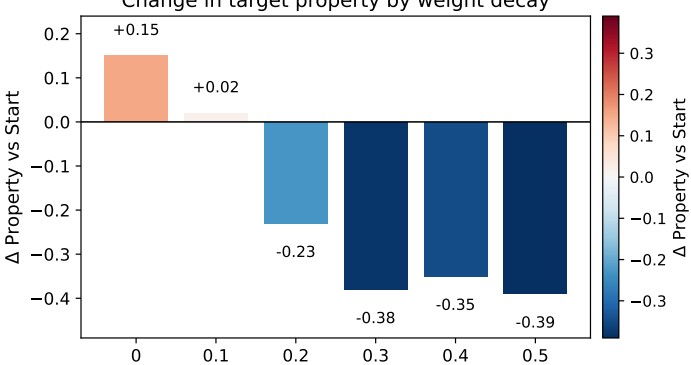

Figure 9: Ablation of the weight decay strength $\lambda$ in the latent-space optimization algorithm.

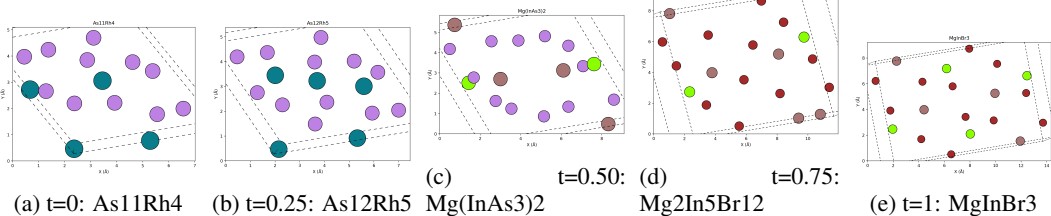

(a) t=0: As11Rh4   (b) t=0.25: As12Rh5   (c)          t=0.50:   (d)          t=0.75:   (e) t=1: MgInBr3
                                          Mg(InAs3)2            Mg2In5Br12

Figure 10: Linear interpolation of the latent representations of As11Rh4 and MgInBr3—2D visualization of primitive cells. The resulting materials gradually change from As11Rh4 into MgInBr3. The changes involve atom count, composition, atom positions, and unit cell shape. This suggests that the latent space offers a smooth way to navigate the transdimensional material space.

## F MATERIAL REPRESENTATIONS

The architecture of our model learns reparameterizations $\mathbf{z}$ of materials M that are meant to navigate the transdimensional space of materials smoothly. To demonstrate their ability to do so, we linearly interpolate two materials, $M^{(0)}$ (As11Rh4) and $M^{(1)}$ (MgInBr3), by linearly mixing their representations $\mathbf{z}^{(0)}$ and $\mathbf{z}^{(1)}$,

$$\mathbf{z}^{(t)} = (1 - t) \cdot \mathbf{z}^{(0)} + t \cdot \mathbf{z}^{(1)}.$$

We then decode the mixed representations and visualize the induced structures in Figure (10). The resulting materials gradually evolve from As11Rh4 into MgInBr3 by altering their composition, unit cell shape, and atom positions, supporting the claim that the latent space smoothly navigates the material space. Such a result justifies an ablation verifying if one can obtain well-performing (in terms of $f(M)$) materials by "mixing" existing structures. To quantify how promising that is, we sample $N_{\text{pair}} = 10$ pairs of existing materials and conduct the linear interpolation, at the same time evaluating $f(M)$ of the decoded, mixed structres.

The results in Figure (11) show that the target property, unfortunately, tends to be higher along the interpolation trajectory. In particular, it reaches its apex in the middle of interpolation. Nevertheless, the failure of this naïve approach strengthens the motivation for more sophisticated tools, like MBO.

Since the latent space of CliqueFlowmer was trained to follow a clique decomposition, we study how individual pieces of this structure impact the represented materials. Thus, we investigate how changes in individual cliques affect the observed material by interpolating a single clique between two examples, while keeping the rest of the representation fixed at one of the interpolated

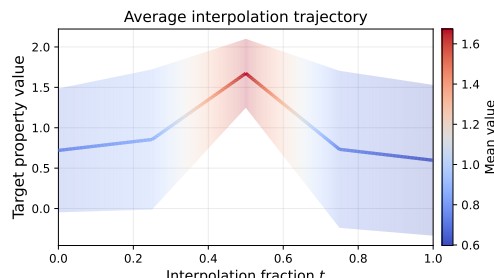

Figure 11: Average $f(M)$ over the course of linear interpolation. The value tends to be higher along the interpolation trajectory.

examples. That is, we fix two latent vectors, $\mathbf{z}^{(0)}$ and $\mathbf{z}^{(1)}$, and change only the coordinates of $\mathbf{z}^{(0)}$ that correspond to the $c$th clique,

$$\mathbf{Z}_c^{(t)} = (1 - t) \cdot \mathbf{Z}_c^{(0)} + t \cdot \mathbf{Z}_c^{(1)}.$$

We visualize this analysis, for materials As11Rh4 and Mg(Cu4Hg)3, and cliques 1 and 3 (out of 8), in Figure (12). The results reveal that these two cliques affect the material structure differently. In particular, alternating clique 1 (top row) does not affect the material decomposition much—on the other hand, it "lifts" two side Rh atoms away from the bottom Rh atom. On the contrary, interpolating clique 3 (bottom row) modifies the position of the four Rh atoms slowly, but it does eventually substitute them for Hg atoms. Ultimately, it completely changes the material composition. This investigation supports a hypothesis that the learned cliques represent different properties of the materials' structure. Formalizing this relationship is not, however, the main focus of this paper, so we leave it to future work.

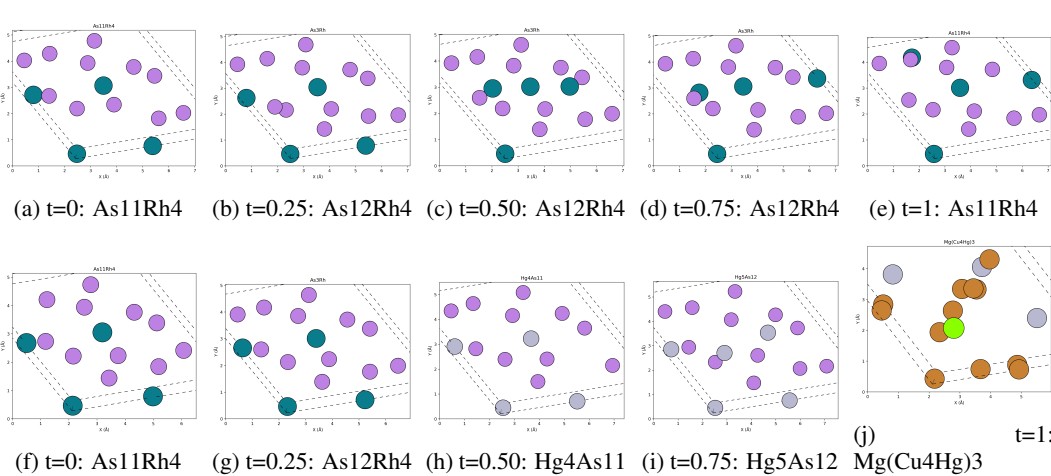

(a) t=0: As11Rh4  (b) t=0.25: As12Rh4  (c) t=0.50: As12Rh4  (d) t=0.75: As12Rh4  (e) t=1: As11Rh4

(f) t=0: As11Rh4  (g) t=0.25: As12Rh4  (h) t=0.50: Hg4As11  (i) t=0.75: Hg5As12  (j)         t=1: Mg(Cu4Hg)3

Figure 12: Linear interpolation of cliques between As11Rh4 and MgInBr3—2D visualization of primitive cells. *Top:* The studied clique does not significantly affect composition or the unit cell shape, but it does affect atom position. In particular, it affects positions of two of the four Rh atoms. *Bottom:* The clique does not alter the material geometry much, but it does significantly affect its composition.

## G  MODEL AND OPTIMIZATION HYPERPARAMETERS

Table 3 summarizes the hyperparameters used throughout all experiments. Unless otherwise stated, these values are fixed across runs.

Table 3: CliqueFlowmer hyperparameters.

| Category | Parameter | Value | Description |
|---|---|---|---|
| Model | n_cliques | 8 | Number of latent cliques |
| | clique_dim | 16 | Dimensionality of each clique |
| | knot_dim | 1 | Overlap between adjacent cliques |
| | transformer_dim | 256 | Transformer hidden dimension |
| | n_blocks | 4 | Transformer layers |
| | n_heads | 4 | Attention heads |
| | n_registers | 2 | Register tokens |
| | mlp_dim | 128 | MLP hidden dimension |
| | n_mlp | 2 | MLP depth |
| | dropout_rate | 0.1 | Dropout probability |
| Loss | alpha_vae | $10^{-3}$ | KL regularization limit |
| | alpha_mse | 1 | Prediction loss weight limit |
| | beta_mse | $10^{-4}$ | Prediction loss weight init |
| | temp_atom | 1 | Atom weight in the loss |
| | temp_flow | 16 | Position weight in flow loss |
| | warmup | $10^5$ | Linear warmup steps |
| | lr | $1.4 \times 10^{-4}$ | Model learning rate |
| Optimization | learner.lr | $3 \times 10^{-4}$ | Latent optimization LR |
| | design_steps | 2000 | Latent optimization steps |
| | decay | 0.5 | Weight decay |
| Data | task | MP-20 | Dataset |

Additionally, for the band gap task, we defined the target property to be the formation energy-regularized band gap

$$f(\mathsf{M}) = \Delta_{\mathrm{band}}(\mathsf{M}) + \lambda_{\mathrm{reg}} \cdot \max\big(0, \mathcal{E}_{\mathrm{form}}(\mathsf{M}) - \tau_{\mathrm{reg}}\big),$$

where we set $\lambda_{\mathrm{reg}} = 20$ and $\tau_{\mathrm{reg}} = -0.2$. The goal of adding the term $\lambda_{\mathrm{reg}} \cdot \max\big(0, \mathcal{E}_{\mathrm{form}}(\mathsf{M}) - \tau_{\mathrm{reg}}\big)$ was to constrain the model's search space to low formation energy materials, as such tend to be more stable. Below this threshold, the objective would simply reduce to the band gap, and the penalty would only be induced for materials that exceed it.

