# OpenReview forum: "Optimizing Materials With CliqueFlowmer"
_ICLR.cc/2026/Workshop/FM4Science — ICLR 2026 Workshop FM4Science Poster_

### Official Review · Reviewer_JBxA · 2026-02-18
**An interesting work of designing and discovering materials in the latent space**

**Rating:** 7
**Confidence:** 3

**Review:**

This paper introduces CliqueFlowmer, a novel framework that applies offline model-based optimization (MBO) to the domain of material discovery (MD), which is challenging but still very useful and promising. By encoding complex, transdimensional material data into a structured latent space, the authors enable the direct prediction and optimization of physical properties through a process of "stitching" high-performing material components together. Their results also demonstrate that CliqueFlowmer outperforms baseline methods.

Strength:
1. Novel methods of material optimization, based on offline MBO, instead of generative modeling.
2. New insights on operating inside the hidden space and utilizing them for optimization and prediction.
3. Proposed a set of methods for a robust optimization strategy, by using Evolution Strategies (ES) with weight decay to keep the optimization within the training distribution.
4. Verified design performance improvement compared to baseline models.
5. Overall, this paper is well-written, and the structure is easy to follow.

Questions:
1. The authors use the S.U.N rate to check if the proposed materials are actually useful, which may not be effective, considering the many efforts made for the optimization. Is there a way to include these S.U.N rate in the design process so that we can always get useful material?
2. The author mentioned that the model is extremely difficult to construct and train. Please also add some discussion on the efficiency and cost of this proposed model compared to other baselines.
3. How would these methods be applied to design for other material properties?

---

### Official Review · Reviewer_FGXd · 2026-02-18
**CliqueFlowmer**

**Rating:** 8
**Confidence:** 3

**Review:**

- Summary
The authors introduce CliqueFlowmer, a model for material discovery leveraging flow matching for geometric features and a causal transformer for producing atom identities. The authors also display CliqueFlower's capability for model-based optimization to further improve performance on evaluation metrics.
- Contributions
1) Introduction to materials design and MBO using flow-matching.
- Pros
1) Differentiated generation of geometric features and atom identities using two separate generative models. Causal atom identity generation is conditioned on the generated backbone.
2) Gradient-based search can be performed given the fixed-dimensional latent space.
- Cons
1) An evaluation of sampling/inference speed compared to other works would be appreciated.
- Decision
This paper presents a novel approach for an underrepresented problem, with fairly rigorous detail. Accept.

---

### Meta-Review · Area_Chair_Y6sq · 2026-02-27

**Recommendation:** Accept (Poster)
**Confidence:** 3

**Metareview:**

This paper proposes CliqueFlowmer, a structured latent-variable model that integrates autoregressive atom generation, flow-matching geometry decoding, and clique-based offline model-based optimization (MBO) for material discovery. Reviewers appreciate the structured design, optimization robustness mechanisms (e.g., ES with weight decay to remain in-distribution), and empirical gains over baselines, though they note missing analysis of sampling speed, computational cost, and broader applicability to other properties or constraints. Overall, this is a decent engineering contribution to offline MBO for materials.

---

### Decision · Program_Chairs · 2026-03-03

Accept (Poster)